# PCRL: Reinforcement Learning Based on Priority Conventions for Microscopically Sequenceable Multi-agent Problems

## Abstract

Reinforcement learning (RL) has played an important role in tackling the decision problems emerging from agent fields. However, RL still has challenges in tackling multi-agent large-discrete-action-space (LDAS) problems, possibly resulting from large agent numbers. At each decision step, a multi-agent LDAS problem is often faced with an unaffordable number of candidate actions. Existing work has mainly tackled these challenges utilizing indirect approaches such as continuation relaxation and sub-sampling, which may lack solution quality guarantees from continuation to discretization. In this work, we propose to embed agreed priority conventions into reinforcement learning (PCRL) to directly tackle the microscopically sequenceable multi-agent LDAS problems. Priority conventions include position-based agent priority to break symmetries and prescribed action priority to break ties. In a microscopically sequenceable multi-agent problem, the centralized planner, at each decision step of the whole system, generates an action vector (each component of the vector is for an agent and is generated in a micro-step) by considering the conventions. The action vector is generated sequentially when microscopically viewed, and such generation will not miss the optimal action vector, and can help RL's exploitation around the lexicographic-smallest optimal action vector. Proper learning schemes and action-selection schemes have been designed to make the embedding reality. The effectiveness and superiority of PCRL have been validated by experiments on multi-agent applications, including the multi-agent complete coverage planning application (involving up to $4^{18} > 6.8 \times 10^{10}$ candidate actions at each decision step) and the cooperative pong game (state-based and pixel-based, respectively), showing PCRL's LDAS dealing ability and high optimality-finding ability than the joint-action RL methods and heuristic algorithms.

## 1 Introduction

**Backgrounds** Multi-agent systems have been applied to various civil (see Fig. 1), military and entertainment applications [1, 2]. With the applications becoming broader and deeper, the agent number in a multi-agent system has become larger and larger [3, 4], which results in the large discrete action space (LDAS) problem. A multi-agent problem with even a dozen of agents is hard to be solved by reinforcement learning (RL) [5] for its explosive joint-action combinations [6].

To tackle the multi-agent LDAS problems, some researchers resorted to independent reinforcement learning regardless of other agents, but that may form loose cooperation and yield unstable or sub-optimal solutions; some researchers resorted to distributed reinforcement learning and communication to cooperate , but that lacks a global knowledge and is hard to design cooperation mechanisms [7];

Submitted to 36th Conference on Neural Information Processing Systems (NeurIPS 2022). Do not distribute.

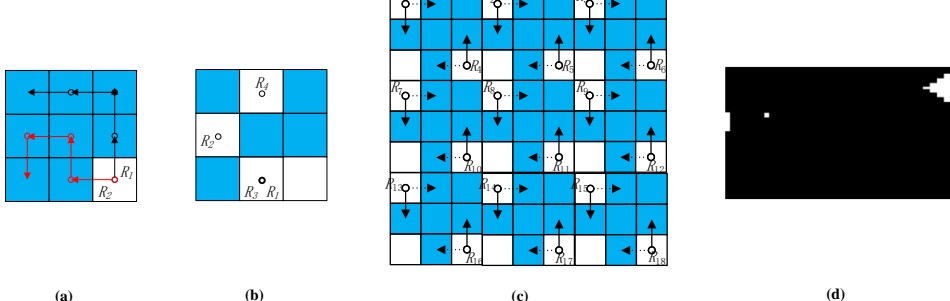

(a)      (b)      (c)      (d)

Figure 1: Examples of microscopically sequenceable multi-agent applications: (a)∼(c) for multi-agent complete coverage path planning (CCPP), where the cleaning agents cooperate to visit the shaded cells in the fewest steps (min-max); and (d) for cooperative pong (CP). (a) A smaller-scale situation needs at least 4 steps for the two-agent system to cover all uncovered cells. The action vectors (i.e., joint-actions) (left,up) and (up,left) for $R_1$, $R_2$ will be tie optimal at current state, but (up,left) is the smallest lexicographic optimal action vector according to our priority conventions. (b) The priorities (can be relative-position determined) for the agents from high to low, at this state, are $R_4 \prec R_2 \prec R_1 \prec R_3$. The state representation includes map state and agent state, and can be "10101110001010020" where the '2' means 2 agents. (c) At this decision step, the system has $|\mathcal{A}^N| = 4^{18} > 6.8 \times 10^{10}$ feasible action vectors ($|\mathcal{A}|$ is the action set size for one agent and $N$ is the number of agents), where the solid and dashed arrows are two tie optimal action vectors among many others. The PCRL will learn to select a specific optimal action vector regarding priority conventions. (d) Cooperative Pong (detailed descriptions see the experiment section). CP can be established as a microscopically sequenceable problem.

many other researchers [8, 9] turned to centralized RL for high performance. For a centralized RL method, a trivial joint-action representation (Fig. 2(a)) can be applied to a small-scale multi-agent system. However, when the agent number increases to a dozen, it becomes unaffordable because the action space is exponential to the number of agents (see Fig. 1(a)∼(c)). Previous centralized RL [8] has tried relaxing large discrete action space to continuous action space and generating a few discrete samples near the optimal continuous action vector to find the locally optimal discrete action vector. Nevertheless, this subspace may miss the optimal discrete action vector thus leading to sub-optimal; Moreover, the relaxing methods lack solution quality guarantees from continuation to discretization.

Therefore, it remains challenging for a centralized RL to directly, effectively and efficiently tackle the multi-agent LDAS problems, difficult in representing the actions and finding the optimal action from $10^{10}$ or more candidate action vectors. Moreover, a multi-agent LDAS problem often has many tie optimal actions; see Fig. 1 for an example. With multiple ties, RL for LDAS is also difficult to train because multiple return-maximum peak landscapes may impact the convergence and exploitation of RL, so new action selection and training schemes are required.

**Motivations**    Some multi-agent problems are microscopically sequenceable, i.e., an optimal joint action can be selected out one component by one component in micro-steps according to the conventions, besides of being selected out simultaneously, i.e., the "microscopically sequenceable" defined in RL language is that the joint action $\pi^*(s) = (a_1, a_2, ..., a_n)$ where each component can be produced as $a_1 = \pi_1^*(s), a_2 = \pi_2^*(s|a1), a_3 = \pi_3^*(s|a2), ...$ and $\pi_i$ functions are related. For example, in Fig. 1(b), the planner (can be set to the smallest id agent $R_1$, without loss of generalization) selects the action for $R_4$, say "right"; then, the planner selects the action for $R_2$ by referring to $R_4$'s "right", and selects out "up"; Then selects out $R_1$ to "up" and $R_3$ to "left". After the planner decides the action for each agent, the planner sends the action to corresponding agents and these agents simultaneously execute the joint action.

The decision flow at a decision step has been shown in Fig. 2(c) and can be expressed by natural language (formal description in later paragraphs) as follows: (1) The planner makes clear the order of agents. (2)The planner decides one action (from $\mathcal{A}$) for the highest priority agent that the planner believes to produce the largest $Q$ value for the whole system, with the following agents fully supporting the highest priority agent. And if the first prioritized agent has tie actions, the action priority will be obeyed. (3) The planner refers to the first prioritized agent's action decision and then decides the action for the second prioritized agent, also aiming for the largest $Q$ value for the whole system. The procedure continues until the last agents.

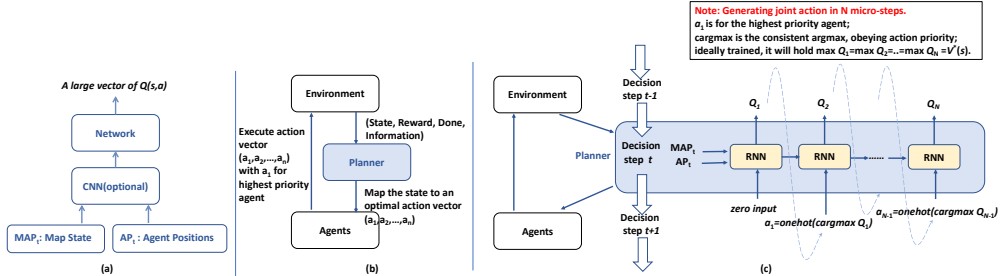

Figure 2: The diagram explanation of methods. (a) trivial joint-action RL for multi-agent LDAS problems, the length of $Q$ will be $|\mathcal{A}|^N$. (b) *macroscopic* view of PCRL. The planner maps the current state (including map state + agent state) to an optimal action vector whose first component is for the highest priority agent, etc. (c) **microscopic** view of PCRL: at this decision step (e.g., Decision step t), the planner outputs in $N$ micro-steps the best action for the highest priority agent and then sequentially for other agents. The length of $Q_i$ is $|\mathcal{A}|$.

**Priority conventions**   The agreed priority conventions for a specific multi-agent problem are pre-scribed by humans before the learning starts, and can be problem-specific designed. The priority conventions usually consist of (1) **Agent priority convention**: This is for determining the order of agents to decide action and breaking symmetries: Dynamically, at each decision step of the centralized system, the agents' priorities are determined by current states, say, the relative positions of agents. For example, in Fig. 1(c), the agents' priorities can be determined from left to right and upper to lower position, so the priorities (deciding order) of agents are $R_1 \prec R_2 ... \prec R_{16}$. Here, $x \prec y$ means $x$ has a higher priority than $y$. If the priority is not position based but ID based, then the state space will have many mirror states. (2) **Agent ID convention**: This is for coinciding agents and breaking symmetries. Statically assign each agent a unique ID before learning to identify itself from others. When at least two agents coincide in the same position during the application, the smaller id agent has a higher priority. (3) **Action priority convention**: This is for breaking ties of equally optimal actions. The actions of an agent have prescribed priority, say, up action$\prec$left action$\prec$down action$\prec$right action, to break action ties if two actions both will bring about optimality, and to converge to one minimum lexicographic action vector. Hereafter, we define that two action vector $\vec{\alpha} \prec \vec{\beta}$ (or called "$\vec{\alpha}$ is **lexicographic smaller** than $\vec{\beta}$") if and only if $\exists i \in [1, N]$, $\alpha_i \prec \beta_i$ and $\forall j \in [1, i-1]$, $\alpha_j = \beta_j$.

The contribution of this paper lies in (1) Priority conventions and proofs: this paper proposes the concept of "microscopically sequenceable" to tackle LDAS multi-agent problems without missing the optimality, and can help exploitation around the lexicographic-smallest optimal action vector. (2) New schemes: it proposes new learning schemes that fully exploit the agreed conventions, such as auxiliary equality constraint, and neural network action selection schemes for PCRL. (3) Proof-of-concept practices: applying PCRL to tackle seemingly different problems, including $10^{10}$ magnitude LDAS multi-agent path planning problems and the pixel cooperative task, and achieves 20% fewer steps and competitive performance.

## 2   Related Work

Nowadays, deep reinforcement learning has been successful in large state problems [5, 10]. However, it is still very challenging to deal with the LDAS problems, especially in the action representation and selection.

### 2.1   Action representation

For the action representation, previous work has followed several lines to circumvent the large-discrete-action-space obstructions:

One line is to parameterize actions into continuous spaces and then to find discrete actions near the optimal continuous action [11] because continuous actions are differentiable. For example, [8]

exploited the k-nearest-neighbor to find the optimal policy $\pi^*(s) \approx \underset{a \in top_k(|a-\hat{a}|_2)}{\arg\max} Q(s, a)$ in a discrete subspace near the optimal continuous actions $\hat{a}$. It has been validated in discrete cartpole problems, recommending systems and multi-agent systems. One step further, work [12] exploitedly utilized the action space structure and without assuming the structure is provided a priori. This line will likely miss the optimal discrete solutions for inappropriate continuation and will lead to suboptimality. Also, it lacks a quality guarantee from continuation to discretization, which also has been encountered and studied in the randomized rounding theory of integer programming [13].

Another line is to factor or eliminate action spaces [14–16], using binary coding and coarse coding to eliminate actions that are not optimal with high probability. However, the elimination probability is hard to model and eliminations are not always possible. Some researchers employed new factor and combination methods [17–20] for action representation and selection, when the problem satisfies Individual-Global Maximization (IGM) Constraint or monotonic properties.

## 2.2 Action selection

Action selection usually has deep entanglement with action representation. [21–23] uses DQN for action selection and has achieved success in Atari games and human-level control. [24] applied a recurrent neural network (RNN) to select actions for POMDP. [25] used bidirectional RNN to represent the actions and select actions, successfully dealing with the continuous multi-actions and have achieved success in SC2 problem and other problems. Some work resorted to social conventions for selecting actions but not for LDAS problems, for example, [26, 27]. And some work relied on sophisticated communications for exchanging selected action [28–30], which may be vulnerable to be impacted by unstable communication in practical cooperation/coordination.

Our way of action selection resembles Stackelberg model and the sequence-to-sequence paradigm (for example image-to-caption [31]). [32] also proposed seq-to-seq and on-policy policy gradient to tackle LDAS problems, along with others [33] but we offer a large agent number off-policy and optimality-preserved one. Meanfield reinforcement learning [4] also has LDAS dealing ability. It iteratively learns each agent's best response to the mean effect from its neighbors and is proved to converge to Nash Q-value. However, the Nash Q-value is not always the global optimal value in scenarios like the prisoner's dilemma.

# 3 Methodology

This section introduces the basics of PCRL, then the mathematics and details of how to learn the agreed priority conventions in PCRL.

**PCRL algorithm** Our concern problems are markovian when *macroscopically* viewed and can be established as a classic MDP $M = \langle \mathcal{S}, A, \mathcal{P}, \mathcal{R}, \gamma \rangle$, where the letters are as usually defined while $A = \mathcal{A}^N$ where $N$ is the agent number and $\mathcal{A}$ is the action set of agent 1 (the studied agents are homogeneous and the action set of each agent is the same as agent 1). PCRL aims to learn the optimal policies to maximize the accumulated discounted return of an MDP.

Without losing generality, we suppose agents $R_1, R_2, ..., R_N$ are priority-ordered according to the agreed conventions, where $N$ is the number of agent. Corresponding to the above basics and Fig. 2, the overall pseudocode of PCRL is in Algorithm 1 in appendix. It is modified from the DQN [5], differing in aspects of action representation-selection, neural network architecture and learning schemes.

**State representation** When *macroscopically* viewed as in Fig. 2(b), the planner maps current state $s$ (consisting of features of current physical map state and agent state) to an action vector:

$$\vec{a_t} = f(s_t), \tag{1}$$

where $f$ is a trainable RNN-based function and is to be optimized. For cleanness, we will omit the subscription of $t$ for the current state hereafter.

The state $s$ can be convoluted features or simply the concatenation of raw map state and agent state, i.e., $s_t = concat(MAP_t, AP_t)$. Using the CCPP (in Fig. 1(b)) as an example, $s =$

$S_{grid\_map}|S_{agent\_positions}$, where "|" means concatenation; grid_map can be binary vector indicating whether each grid has been covered (i.e., visited), and agent_position are an integer vector with length equal to grid_map indicating the agent number upon each cell.

For simplicity, the state $s$ will be supplied as the initial hidden state (other usages are also possible), i.e., the $h$ and $c$, of the RNN of the planner.

**Action representation and selection**    If the action space is not large enough, then a joint-action representation and selection will work. However, large action brings in intractability. The action in PCRL is represented using recurrent neural networks. When generating $\vec{a} = (a_1, a_2, ..., a_n)$, PCRL resembles the image-to-caption [31] technology in that when given a state $s$, PCRL learns to optimize and produces a sequence of $a_i$, $i \in [1, N]$ with $a_1$ assigned to the highest priority agent and this sequence aims to bring about the maximum return. Mathematically, the action for the highest priority agent (i.e., the $a_1$) is generated as below:

$$a_1 = \text{argmax } RNN((h_0, c_0), \vec{0}) \triangleq \text{argmax } \overrightarrow{Q_1}, \tag{2}$$

where $h_0 = c_0$ is the feature of state $s$, and supplied as the initial state of the RNN. $\vec{0}$ is the input of the RNN in this micro-step and means no agent has been decided an action (can refer to Fig. 2(c)). cargmax (stands for consistent argmax) function means selecting the action for the highest priority agent, i.e., $R_1$ according to the agreed action priority convention if ties exist. $RNN()$ function is a one-step forward function of the RNN, which is with only $|\mathcal{A}|$ output entries. If ties exist in output entries, argmax should consistently select the smallest optimal action according to action priority, so hereafter we write argmax as cargmax . For the cargmax function, say, if $RNN()$ outputs a value vector of $[5.1, 5.32, 5.2, 5.32]$, then we consistently select the first "5.32" (index=1) in case of ties whereas trivial argmax in pytorch may return unguaranteed and inconsistent index.

*Remark 1: Better comprehended with CCPP in Fig. 1(b). $\overrightarrow{Q_1}$ means the expected Q value for the whole system when agent 1 selects each action from $\mathcal{A}$, so $\overrightarrow{Q_1}[j]$, $j \in |\mathcal{A}|$ tries to approximate*

$$\max_{a_2 \times a_3 \cdots \times a_N \in A^{N-1}} Q(s, (a_1 = action_j, a_2, \cdots, a_N)) \triangleq \overrightarrow{Q_1^*}[j]. \tag{3}$$

*This approximation is possible since $s$ contains all information of the current state including the total number of agents, and the following agents refer to and fully cooperate $a_1 = action_j$. (Eq. 2) can also be viewed from state mapping, i.e., $\overrightarrow{Q_1}$: (grid states, agent state, total=N, priority=1 (or highest)) $\rightarrow$ 4 real values $\rightarrow$ one index. This mapping can be learned by reinforcement learning.*

Similarly, for the remaining agents, the planner conditions at previous agents' actions and the priority conventions as well, so

$$a_i = cargmax \ RNN((h_{i-1}, c_{i-1}), onehot(a_{i-1})), \tag{4}$$

where $onehot$ function maps an integer into a 0-1 input in which there is only one 1 in index $a_{i-1}$.

*Remark 2: Assume now $i = 2$, then now this $RNN(.)$ ($\triangleq \overrightarrow{Q_2}$)outputs $\mathcal{A}$ entries and approximates to*

$$\max_{a_3 \times a_4 \cdots \times a_N \in A^{N-2}} Q(s, (a_1, a_2 = action_j, a_2, \cdots, a_N)) \triangleq \overrightarrow{Q_2^*}[j], \tag{5}$$

*where $a_1$ has been determined at previous micro-step and conditioned at this micro-step.*

The approximation in each micro-step has physical meaning: If sufficiently trained, then

$$\overrightarrow{Q_1^*} \triangleq RNN^*(s, \vec{0}), \ len(\overrightarrow{Q_1^*}) = |\mathcal{A}| \tag{6}$$

$$V^*(s) = \max_{a_1 \in \mathcal{A}} \overrightarrow{Q_1}^* \tag{7}$$

$$\overrightarrow{Q_2^*} \triangleq RNN^*(s, (\text{cargmax}\overrightarrow{Q_1})), \ len(\overrightarrow{Q_2^*}) = |\mathcal{A}| \tag{8}$$

$$V^*(s) = \max_{a_2 \in \mathcal{A}} \overrightarrow{Q_2^*} \tag{9}$$

$$\cdots \cdots \tag{10}$$

$$\max \overrightarrow{Q_1^*} = \max \overrightarrow{Q_2^*} = ... = \max \overrightarrow{Q_n^*}. \tag{11}$$

where function $len$ is the number of output entries. The physical meaning can be comprehended as that the RNN is to approximate the optimal value of the monothetic system. The last line is important because $V^*(s)$ links all the $\overrightarrow{Q_i^*}$ and can be exploited to achieve better learning.

**Reward**  The reward at each step is the total reward that all agents earn. The total reward is beneficial for learning and regards the system as a monolithic system. In Fig. 1(c), the reward is the number of uncovered grids that the whole system visits at this step. An uncovered grid will give a reward of 1 only for the first time it was visited. If $R_1 \sim R_{16}$ executed the solid arrow action, the reward would be 16. Sometimes, a step penalty is added to the step reward so that the hypothesis reward [34] in RL holds[1], so that, maximizing the return of the MDP $\leftrightarrow$ minimizing the step to cover all shaded cells.

**Training**  As Fig. 2 and Algorithm 1 depict, the training strategy needs specially designation. When in state $s$ and takes action $\vec{a}$, the MDP will transfer to some $s'$ and get a reward $r$, then $Q(s, \vec{a})$ should converge to the target $Q_{target}$,

$$Q_{target} = r + \gamma \max_{\vec{a'}} \hat{Q}(s', \vec{a'}) \tag{12}$$

where $\vec{a'}$ can be selected out on the target network according to (Eq. 2) to (Eq. 4), rather than enumerating all $\mathcal{A}^N$ action combinations. The $Q(s, \vec{a}) = Q(s, (a_1, a_2, ..., a_N)$ can be calculated by feeding $\vec{0}, onehot(a_1), ..., onehot(a_{N-1})$ to the RNN in the $N$ micro-steps and gathering the $a_N$-th entry in $\overrightarrow{Q_N}$ (see Fig. 2(c)).

Moreover, considering the uniqueness of our PCRL decision flow and (Eq. 11), we can use an auxiliary loss to accelerate and smoothen the training. The auxiliary loss is equalities among agents' $Q_i$ values (as explained in (Eq. 11) and that paragraph), i.e.,

$$L_{aux}(ss, \overrightarrow{aa}, rr, ss) = (\max Q(ss, \vec{0}) - Q_{target})^2 + (\max Q(ss, (\vec{0}, onehot(aa[1]))) - Q_{target})^2$$
$$+ \cdots + (\max Q(ss, (\vec{0}, onehot(aa[1]), ..., onehot(aa[N-1]))) - Q_{target})^2, \tag{13}$$

where $ss$ and $\overrightarrow{aa}$ are batch data from the experience replay buffer. $L_{aux}$ guides the mean-squared error of $Q$s of each agents to converge to target $Q_{target}$. Other type of (Eq. 13) is also possible.

We propose the following, and the proofs are in the appendix.

**Theorem 1.** *The action representation scheme and selection process will not lose the optimality of the action space and can select out the lexicographic-smallest optimal action vector.*

**Theorem 2.** *The action selection process breaks the exponential action space into linearly expressible space complexity.*

# 4   Experiments and Discussions

We evaluate the performance of PCRL on two seemingly different but both can be viewed as microscopically sequenceable multi-agent applications: multi-agent complete coverage path planning (CCPP) problem and Cooperative Pong (CP). We compared our method with joint-action RL or heuristics on their efficacy and efficiency. All experiments were carried out on a computer of amd ryzen 9 5950x, with a single 3090 RTX GPU. Every method has run 10 times with different random seeds at each experiment setting and reports the averages and the standard deviations (STD), to gain the reliability of the evaluation. The hyperparameters are listed in the appendix, also in the source code that is supplemented.

Our experiments consist of (1) The PCRL's LDAS dealing ability and optimality-finding ability on CCPP, compared to joint-action DQN. (2) the PCRL's optimality-finding ability on CCPP, compared to the heuristic algorithm. (3) the PCRL's optimality-finding and end-to-end ability on CP, compared to joint-action DQN.

---

[1]http://incompleteideas.net/rlai.cs.ualberta.ca/RLAI/rewardhypothesis.html

### 4.1 Multi-agent complete coverage planning

**PCRL illustration and consistent argmax** Table 1 depicts the sufficiently trained (for 1 hour) and converged results of the two-agent complete coverage path planning example on the $3 \times 3$ gridworld. The first column is the current state, where "1" stands for shaded grid cell and R stands for a agent. The agents are homogeneous and the priorities are position-determined, so we ignore the ID of them. The second column shows the outputs of the RNN (i.e., the Q values of RL) in two micro-steps. The first micro-step has the Q value for the highest priority agent. It is the expectation for the first agent to decide an action and the following agent to decide a corresponding action . And the second micro-step is the second agent's Q value referring to the first agent's action and fully supporting the first agent. As can be seen, the bolded values are near, which is almost the $V^*(s)$ (The ground truth $V^*(s)$ at each current state can be easily calculated).We can see that the total return received in the first state is six, which is eight plus the penalty multiply minimal steps. The third column is the optimal action for each agent, obeying to the priority. The fourth column is the reward for the whole two-agent system. The last line column is some explanation for data interpretation. Due to the cargmax function, this RNN has converged to the smallest lexicographic action vector at each state, and in training stage helped exploitation.

Table 1: CCPP illustration of PCRL and consistent argmax

| Current state | Q value at two micro-steps after sufficiently trained | Converged Optimal action | Reward | Explanation |
|---|---|---|---|---|
| 1 1 1
1 1 1
1 1 R,R | [[5.7079, 5.5537, 4.9969, 5.1314],
[4.9332, 5.7386, 5.1010, 5.0878]] | (up, left) | 2+(-0.5)=1.5
(number of newly covered cells
plus the step penalty) | Initital state. The highest priority agent has tie optimal actions (up or left). It selects lexicographic smaller "up". The second agent referring to this selects "left" and can get the whole system the largest Q. |
| 1 1 1
1 1 R
1 R 0 | [[4.2621, 3.5443, 3.3271, 3.6442],
[4.2466, 3.9317, 4.1653, 3.4481]] | (up, up) | 2+(-0.5)=1.5 | At this step, the second priority agent has near "up" or "left" action and "up" is selected. |
| 1 1 R
1 R 0
1 0 0 | [[2.1108, 2.7424, 1.9205, 2.1509],
[2.0065, 2.7412, 2.4293, 2.0047]] | (left, left) | 2+(-0.5)=1.5 | This action vector is far better than other action vectors. |
| 1 R 0
R 0 0
1 0 0 | [[0.6524, 1.5007, 0.3517, 0.4291],
[0.5143, 0.9889, 1.5010, 0.4947]] | (left, down) | 2+(-0.5)=1.5
(ground truth $V^*(s)$ value quite close to this rewards) | After executing (left, down), the mission is completed (done). |

**PCRL and DQN in deterministic transition environment CCPP** We compared the final performance of PCRL, joint-action DQN on grid world on various numbers of agents. The details of the results are shown in the Fig. 3.

Fig. 3(left) shows that when trained enough steps, PCRL can get a close final performance to joint-action DQN. That means the PCRL network can successfully learn the priority conventions and the proper Q values, and has an ability similar to the monothetic system. Moreover, PCRL has the potential to deal with over $10^{10}$ actions: the needed parameters of DQN are exponentially increasing while PCRL has a linearly increasing number of parameters, about 562176 for 12 agents (see the middle of Fig. 3). The performance of DQN is slightly better than PCRL, however the performance decreases quite rapidly and soon fails to tackle 12 or more agents scenarios (see the right of Fig. 3, where the averaged return of PCRL is normalized as 100%).

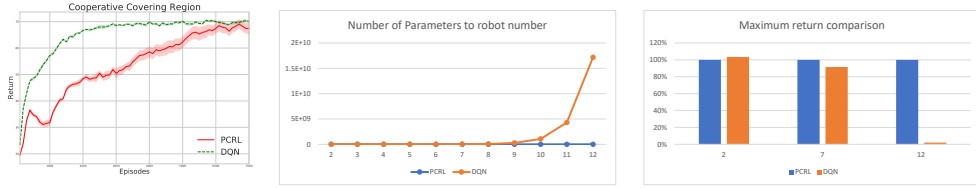

Figure 3: Comparisons of PCRL and joint-action DQN. left: performance on CCPP of 2 agents and $10 \times 10$ map size with 99 cells except the down-rightmost to visit in deterministic transition environment; middle: the total number of needed parameters; right: performance of DQN dives comparing to PCRL's.

**Large space and non-deterministic transition environment** The Fig. 4 depicts 18 agents on a $9 \times 9$ grid world the same as Fig. 1(c), with 54 uncovered cells in total. This case, at each step, owns over $10^{10}$ action vector candidates. The experiment is further made tougher to a non-deterministic (stochastic) environment. In the non-deterministic transition environment experiment, there is a non-zero probability $p$ of "leftwards slipping" when agents take actions (say agent takes "up" then it has $p$ probability being moved to the "left" cell), where agents' do not have any prior knowledge about

the environment transition. Fig. 4 shows the averaged score and averaged steps with sliding-window width 100 in the training episode. As shown, the training is converging gradually and smoothly. The averaged steps is approaching 20 and the averaged return is about 44.

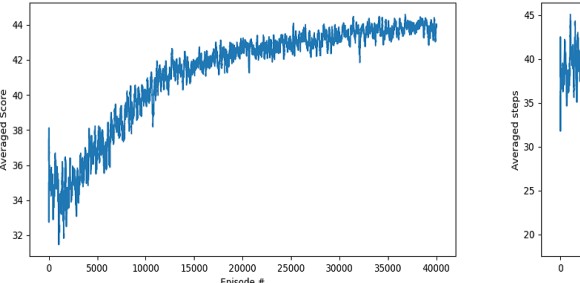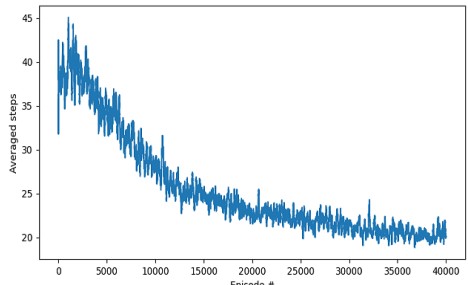

Figure 4: The training of 18 agents in the stochastic environment, where the joint-action DQN is unable to tackle.

To further compare the steps, which is the focus of CCPP, we employ a heuristic search algorithm where the agent selects the minimum Manhattan distance cell and goes towards the cell. The Violin plot is depicted in Fig. 5, where the first "violin" is the performance of RNN in mid-course of training, while the second one is of the final RNN after training. The third one is for the heuristic algorithm. PCRL can learn to optimize and use 20% smaller steps (19.7 VS 24.5) to cover all the shaded cells, and is more stable.

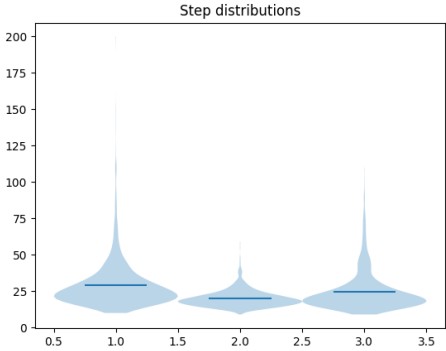

Figure 5: 18 agent performance of PCRL (mid-course and final) compared to the heuristic. Mean/Stdev of steps are 28.9/16, 19.7/5.6, 24.5/13.3, respectively.

## 4.2 Cooperative pong

Cooperative pong Fig. 1(d) is a game where the objective is to keep the ball in play for the longest time. The game is over when the ball goes out of bounds from either the left or right edge of the screen. All collisions of the ball are elastic. Cooperative Pong has 480 pixels $\times$ 280 pixels, so a ball with speed 9 will cost about 60 steps to go from the left paddle to the right paddle. Algorithms that can keep the ball for 400 steps are rare [35]. The CP can be established as a joint-action DQN with and a microscopically sequenceable multi-agent problem. To show the applicability of PCRL in CP task, we applied PCRL on the state-based and pixel-based CP, respectively.

**State-based CP and pixel-based CP** We trained PCRL and the joint-action DQN on CP with a similar scale of network parameters, and the performance is illustrated in Fig. 6a. Set ball_speed=9 pixelsstep, left_paddle_speed=28, right_paddle_speed=28, cake_paddle=False, max_cycles=900, bounce_randomness=False, each frame reward=1, out boundary reward=-100. We can see that PCRL even outperforms the joint-action DQN which treats the two agents as a monothetic agent. The key reason for this result may be that PCRL has higher parameters efficiency than DQN of a similar scale of parameters.

To better demonstrate the effectiveness of our approach, we also use the pixel states as the Q-net inputs. We resize the original pixel state to 84 * 84 * 3 tensor. We use a 3-layer convolution network transforming the pixel state to a vector state first, and then feed the vector state as the RNN hidden state. The performance in training with pixel states (in about four hours) is shown in Fig. 6b, from which we can see that our algorithm converges after about 1000 episodes and get a satisfactory performance. It indicates that PCRL is effective and efficient even in the end-to-end setting without losing much performance compared to the state-based pong performance.

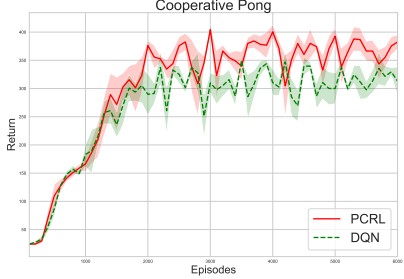 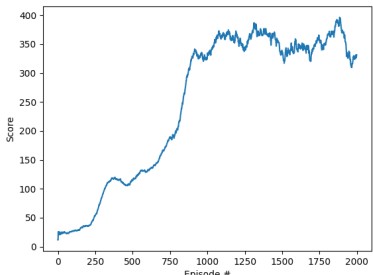

Figure 6: Convergence of $Q_i$. a: Training performance of state-based cooperative pong. b: training of pixel-based CP (playing video attached)

# 5    Conclusion

This paper tackled the microscopically sequenceable multi-agent problems via reinforcement learning that combines agreed priority conventions. The agreed priority conventions can successfully break the exponential scale actions into a sequence of actions (linear scale expressible) without losing optimality. From the multi-agent complete coverage planning and cooperative pong experiments, the algorithm shows closeness or superiority in effectiveness and efficiency to the joint-action DQN and heuristics. The results suggest the potentials of PCRL for a larger number of multi-agent systems (say recommending systems) or swarm robots.

In the future, further work can be done, such as (1) larger agent number multi-agent systems to verify the ability of PCRL for larger action space problems, say swarm robots. (2) application of this algorithm to fully-distributed or centralized-training-and-decentralized-execution multi-agent systems. (3) studies of PCRL for heterogeneous agents. (4) studies on partial observable domains.

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

# A  Appendix

## A.1  Pseudocode of PCRL

---
**Algorithm 1** PCRL: Priority convention RL
---
1: Initialize replay memory $D$; Initialize the evaluation RNN (the action-value function $Q$) with random weights $\theta$; Initialize the target RNN (the action-value function $\hat{Q}$) with weights $\theta^- = \theta$
2: **for** episode=1 to E **do**
3:    Observe initial state $s$
4:    **while** unterminated **do**
5:        Select an action $\vec{a}$: with probability $\varepsilon$ select a random action vector by uniform sampling; otherwise, generate $\vec{a}$ as Fig. 2 and (Eq. 2) to (Eq. 4)
6:        Carry out action $\vec{a}$
7:        Observe reward $r$ and new state $s'$
8:        Store experience $< s, \vec{a}, r, s' >$ into replay memory $D$
9:        Sample random transitions $< ss, \overrightarrow{aa}, rr, ss' >$ from replay memory $D$
10:       Calculate target for each minibatch transition:
          if $ss'$ is terminal state, then $tt = rr$
          otherwise $tt = rr + \gamma * \max_{\vec{a'}} \hat{Q}\left(ss', \vec{a'}\right)$ where $\vec{a'}$ is generated as Fig. 2(c) and (Eq. 2) to (Eq. 4)
11:       Train the evaluation $Q$-network using $(tt - Q(ss, \overrightarrow{aa}))^2 + L_{aux}(ss, \overrightarrow{aa}, rr, ss')$ as loss with respect to $\theta$, where $L_{aux}$ is the auxiliary convergence loss in (Eq. 11) and (Eq. 13)
12:       Update target network as $\theta' = (1 - \tau)\theta' + \tau\theta$
13:       $s = s'$
14:    **end while**
15: **end for**
---

## A.2  Microscopic generation proofs

Theorem: *The action representation scheme and selection process will not lose the optimality of the action space and can select out the lexicographic-smallest optimal action vector.*

*Proof.* Suppose $\vec{\alpha} = (\alpha_1, \alpha_2, ..., \alpha_N)$ and $\vec{\beta} = (\beta_1, \beta_2, ..., \beta_N)$ are two (More than two tie optimal action vectors can also hold similar conclusion.) optimal tie actions for a state $s$, and in the article has deduced that $V^*(s) = Q(s, \vec{\alpha}) = Q(s, \vec{\beta})$.

Since the state $s$ is the same, i.e., the map and agent states are the same, thus the agent priority orders are identical. Suppose $\vec{\alpha}$ is lexicographic smaller than $\vec{\beta}$ (this is possible, since each component of $\vec{\alpha}$ and $\vec{\beta}$ are comparable according to agreed action priority, so either $\vec{\alpha}$ is lexicographic smaller than $\vec{\beta}$ or $\vec{\beta}$ is lexicographic smaller than $\vec{\alpha}$. The latter can prove the following similarly).

Let $j = min_i(\alpha_i \prec \beta_i)$, the planner's procedure will select $\alpha_i$ instead of $\beta_i$ for agent $i$.

Since the smallest-lexicographic action vector can be selected out and learned, PCRL will not lose the optimality for the concerned problem. □

Theorem: *The action selection process breaks the exponential action space into linearly expressible space complexity.*

*Proof.* The recurrent neural network has $N$ micro-step outputs, each micro-step output consists of $|A|$ actions. Suppose that at each micro-step, the planner can select one action, so the possible total action combination space is still $|A|^N$, so RNN will not miss expressing the optimal vector. However, the optimum (maximum) action vector can be expressed by and selected from $|\mathcal{A}|$ per micro-step $\times$ $N$ micro-steps. □

### A.3 Common hyper-parameters

The common hyper-parameters for experiments are:

Table 2: Add caption

| Tasks | CCPP | CP |
|---|---|---|
| learning rate | 1.50E-04 | 1.50E-04 |
| replay buffer size | 2.00E+05 | 2.00E+05 |
| batchsize | 128 | 128 |
| episode length | 200 | 900 |
| $\gamma$ | 1.0 with step penalty 0.5 | 0.995 |
| $\tau$ | 1.00E-03 | 1.00E-03 |
| k(weight update frequency) | 4 | 4 |
| episode number | min(40000,H * W * agent_nums * 50) | 2000 |
| epsilon_start | 1 | 1 |
| epsilon_end | 0.01 | 0.05 |
| epsilon_decay | 0.99992 | 0.95 |
| LSTM hidden size | 1024 | 512 |
| auxiliary loss | type 1 | type 2 |

The auxiliary loss type 1 is as (Eq. 11), and type 2 are the mean squared error of $\max Q_1$ to $\max Q_2$, $\max Q_2$ to $\max Q_3$ etc. This two can get similar results.

The $\gamma$ of CCPP is set as listed because of the reward hypothesis.

Reinforcement learning is based on the reward hypothesis, that is, RL is to maximize $G_t = \sum_{k=0}^{\infty} \gamma^k r_{t+k+1}$. When $t = 0$, it is maximizing $G = \sum_{k=0}^{\infty} \gamma^k r_{k+1} = R_1 + \gamma * r_2 + \gamma^2 * r_3 + ...$, i.e., the maximum expected return from the initial setting of the environment.

If one wants to maximize the reward $G$ to get the fewest step coverage route of the agent, then one needs to set appropriate parameters such as $\gamma$ so that $G$ fits the aim. Generally, $0 \leq \gamma \leq 1$, when $\gamma$ is 0, it is a single-step reflective and myopic behavior. In this appendix, we want to maximize the reward $G$ to obtain the fewest steps of multi-agent complete coverage. Because the multi-agent is treated as a monothetic system, so one agent's proofs can naturally be extended to a multi-agent system. Through mathematical analysis, the following theorems and proofs can be obtained:

**Theorem 3.** *When $\gamma < 1$ and there is no single-step penalty for each step, maximizing the return does not always get the fewest step complete coverage. When $\gamma = 1$ and each step has a single-step penalty (step_penalty<0), maximizing the return is equivalent to the fewest step complete coverage.*

Before the proof, use Fig. 7 to illustrate the first sentence when $\gamma < 1$. The figure shows the same environment configuration (setting), including the same map state and agent position. The two paths of the agent are path (1) and path (2). Denote their rewards are respectively $G^{(1)}, G^{(2)}$. It can be seen in the figure that path (1) is shorter than path (2). However, when $\gamma < 1$ and each step has no penalty, the return of path (2) is larger, that is, $G^{(1)} < G^{(2)}$: According to the return formula, the return for paths (1) and (2) is $G^{(1)} = 1 + \gamma^2 + \gamma^3, G^{(2)} = 1 + \gamma^1 + \gamma^4$, at this time

$$G^{(1)} - G^{(2)} = (1 + \gamma^2 + \gamma^3) - (1 + \gamma^1 + \gamma^4) \tag{14}$$
$$= \gamma^2 + \gamma^3 - \gamma^1 - \gamma^4 \tag{15}$$
$$= (\gamma^2 - \gamma^1) - \gamma^2(\gamma^2 - \gamma^1) \tag{16}$$
$$= (\gamma^2 - \gamma^1)(1 - \gamma^2) < 0 \tag{17}$$

The last step is because when $0 < \gamma < 1, \gamma^2 < \gamma^1$ and $1 > \gamma^2$.

However, When $\gamma = 1$ and a single-step penalty $step\_penalty = -0.2$, the return of path (1) is 2.2, and the return of path (2) is 2.0. This fits the requirements of CCPP: fewest steps and maximum return.

From the illustration of the figure, a mathematical proof of the theorem 3 can be obtained.

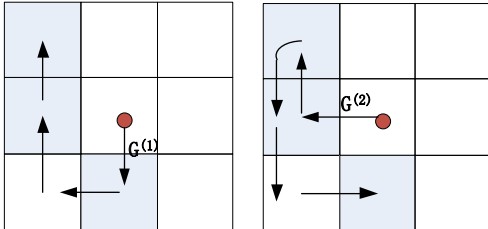

Figure 7: Example diagram of maximum reward hypothesis and shortest path

*Proof.* Because there are examples shown in Fig. 7, the first sentence of the theorem can be proved by contradiction. For the second sentence of the theorem, it can be equivalent to "When $\gamma = 1$, and each step has a single-step penalty (step_penalty<0), a greater return is equivalent to a fewer step complete coverage path".

The idea of the proof is to assume that there are path (1) and path (2), and the number of steps used are $n$ and $m$, respectively. Assume $n < m, n > 0, m > 0$ and $step\_penalty < 0$.

First to prove the necessity, that is, under the same environment configuration, fewer steps will get a greater return, that is, one needs to prove $G^{(1)} > G^{(2)}$. Denote $C_i$ the cell at step $i$ and $C_i = 1$ if the cell has never been visited.

Because $G^{(1)} = (C_1 + step\_penalty) * \gamma^0 + (C_2 + step\_penalty) * \gamma^1 + ... + (C_n + step\_penalty) * \gamma^{n-1} = (C_1 + step\_penalty) + (C_2 + step\_penalty) + ... + (C_n + step\_penalty) = (C_1 + C_2 + ... + C_n) + n * step\_penalty = number\_of\_uncovered\_cells + n * step\_penalty$.

Similarly, $G^{(2)} = (C_1 + C_2 + ... + C_m) + m * step\_penalty = number\_of\_uncovered\_cells + m * step\_penalty$. Due to the penalty of each step, we get $G^{(1)} > G(2)$, that is, a smaller number of steps gets a higher reward.

Proof of adequacy, that is, greater returns can lead to fewer steps. Because the rewards of a scene can be written as $number\_of\_uncovered\_cells + k * step\_penalty$, where $k$ is the number of steps visiting all shaded cells, a larger reward corresponds to a smaller number of steps. □

## A.4 Planning difficulty for CCPP

CCPP can be established as a Euclidean Traveling Salesman Problem which is NP-hard. Here we establish it as an MDP. To realize the difficulty of the problem to set a proper problem scale, we first conducted a agent random wandering experiment to cover all grids and the experimental results are shown in Fig. 8: the $x$-axis represents the $n$ of the square $n \times n$ gridworld, where the $y$-axis is the expected steps (averaged among 1000 runs) for a agent to visit all $n^2 - 1$ cells (the start cell is in down-rightmost and is empty). It can be seen that the number of steps required increases almost exponentially (the log linearly interpolation curve has been fit and has been plotted in the same figure). When $n$ is 10, the number of random steps to completely cover all cells is more than 1000 steps, so we set it as the scale of the experiments. Roughly estimated, the total number of paths for a problem scale of $10 \times 10$ map size and 2 agents can reach 100! (about $10^{158}$). So, this pre-experiment has partially shown the complexity of problems that PCRL can tackle and have strong optimization ability.

## A.5 Deterministic transition environment of 18 agents CCPP

Fig. 9 shows the settings of Fig. 1(c) and with a deterministic transition environment. It shows the return and steps of each training episode, the training cost and the sliding-window averaged steps. The ground truth return is 52.5 and PCRL sometimes can get 51.5 and an average near 50. This picture again shows the LDAS dealing ability and high optimality-finding ability of PCRL.

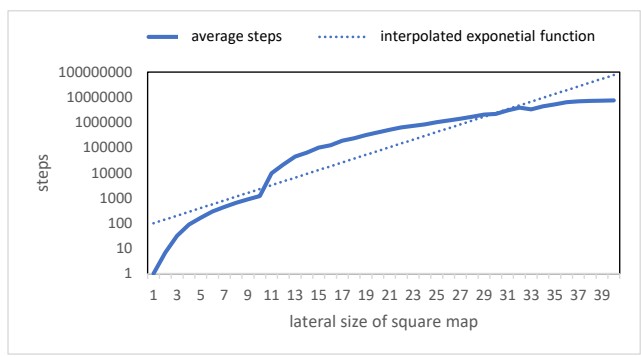

Figure 8: Expected steps to cover all cells with random wandering

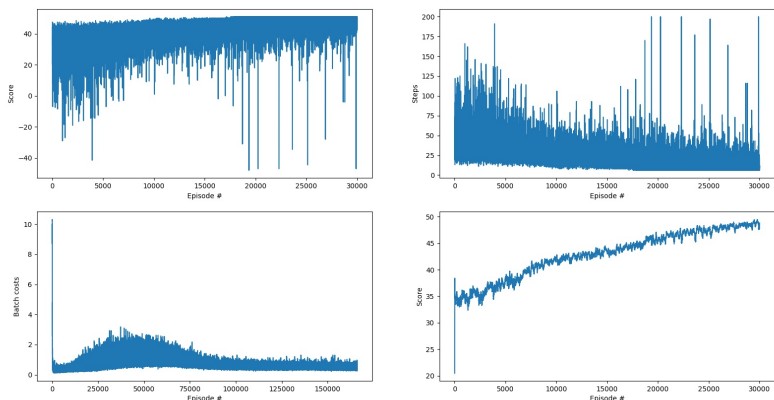

Figure 9: The training of 18 agents in deterministic environment.

## A.6 Gap of max Qi in state-based CP

As shown in (Eq. 11), the maximum of $\overrightarrow{Q_1}$ will equal to the maximum of $\overrightarrow{Q_2}$ at the state $s$, theoretically. Respecting this, we evaluate our trained neural network to analyze the convergence of $Q_i$s. Define the gap as $(\max \overrightarrow{Q_2} - \max \overrightarrow{Q_1})/\max \overrightarrow{Q_1}$. The left part of Fig. 10 shows the gap at every state along one test episode, while the right part shows the averaged gap in the training history of an RNN. The former demonstrates an only 5% gap in the states. The latter demonstrates that as training goes, the gap from substantially large becomes narrow and narrow. Both parts proved the convergence of gaps between $\overrightarrow{Q_i}, i = 1, ..., N$.

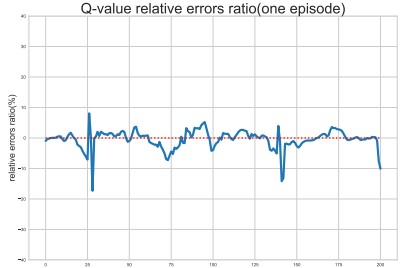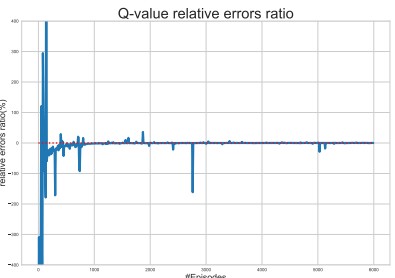

Figure 10: Convergence of $Q_i$. Left: The gap along a test episode. Right: The evolution of gap in the training stage.

