# OpenReview forum: "PCRL: Priority Convention Reinforcement Learning for Microscopically Sequencable Multi-agent Problems"
_NeurIPS.cc/2022/Conference — NeurIPS 2022 Submitted_

### Official Review · Reviewer_3mod · 2022-07-07

**Rating:** 3
**Confidence:** 5
**Soundness:** 1 poor
**Presentation:** 1 poor
**Contribution:** 2 fair

**Summary:**

The authors attempt to tackle the problem of RL in settings with large action spaces caused by there being many agents in the system. The authors provide a mechanism to pick an ordering over which actions for each agent are generated by a planner and show that they can pick the action that is "lexicographically smallest"

**Questions:**

What is the connection between the results in this work and the results in “Multi-Agent Reinforcement Learning is a Sequence Modeling Problem”? and how does this paper go beyond the results there?

What is the importance of tie-braking and the connection to social conventions posited in the introduction?

What is the importance of small lexicographic orderings?


## PCRL: Priority Convention Reinforcement Learning for Microscopically Sequencable Multi-agent Problems

### Grammar + sundry things:

- I think in the title the word should be sequenceable?
- In abstract it should be “existing work has mainly tackled”
- Figure 2 is too low resolution and has to be improved to be legible. This is essential.
- The paper is full of what I would call “unbacked opinions”. The authors will say things like “the probability is hard to model” (line 104). Hard how? Why? The reviewers are asked to believe this on faith rather than with some backing. Or in line 178 the authors say “the approximation has physical meaning” but don’t state what it is.
- The paper should probably have a reference to “**Multi-Agent Reinforcement Learning is a Sequence Modeling Problem”** which contains quite similar ideas.
- What is the relevance of the reward hypothesis in lines 185-186? Also, there’s a paper to cite for this rather than a footnote.

### Writing:

- There are many components in the introduction that need to be rewritten to make the contributions of the paper clearer:
    - What do social conventions have to do with microscopic and macroscopic sequenceable models?
    - Why are priority conventions important?
    - Why is the sequencing a solution to action space blow-up?
    - More intuition on what microscopic sequenceability is
- This paper is not written in standard academic form and employs colloquialisms in many places such as lines 129-130
- There are spelling mistakes through the paper. Please employ a spell-checker.

**Limitations:**

The authors, in the conclusion, point out that they investigate their results on a relatively limited number of systems.

**Strengths And Weaknesses:**

Unfortunately, this paper is written in a manner that makes it not possible to assess the quality of the paper. The main figure in the paper is unreadable, basic definitions are missing, spelling errors are abundant in almost every paragraph, and the figures are low enough pixel density that I have trouble reading them on my laptop (they appear to be screenshots). This is unfortunate because the underlying idea, to the extent that I can figure out what it is, that actions can be selected sequentially in centralized, cooperative multi-agent problems is interesting and the related work suggests the authors have expertise on this topic. I would ask the authors to do a more thorough job in preparing their paper for submission and provide some suggestions on how to do so.

---

> ### Author Response · Authors · 2022-08-02
> **Author response**
>
> Thanks for the reviewer's  comments and questions.
>
> For the "Questions" part:
>
> response to Q1: “Multi-Agent Reinforcement Learning is a Sequence Modeling Problem” was uploaded on 30th, May while our PCRL was submitted on 19th, May, so the authors of both papers were not aware of another paper. This interesting coincidence again showed the significant meaning and novelty of our paper. The authors of two papers independently developed their own idea and experiments.  The difference between our methods and the MAT is that: (1) in PCRL, agents’ order is relative position based, and can reduce the symmetries of the state mapping into joint action. This priority ordering can help exploitation around the smallest lexicographic joint-action. (2) PCRL considered multiple optimal joint-action situations, which are commonly encountered as figure 1 shows. (3) MAT is advantage decomposition based while ours is Q value-based and both have shown the correctness independently. (4) We use LSTM rather than Transformer, with fewer parameters and can the number of parameters increases less slowly. So PCRL is light weighted and scalable. (6) The idea of MAT borrows ideas from the Transformer and largest sequence model, while ours borrowed from the social conventions and partial ordering in discrete mathematics.
>
> response to Q2 and Q3: For example, in  many decomposition based RL, they often requires IGM principle, i.e., $\operatorname{argmax} Q_{\text {total }}(\boldsymbol{s}, \boldsymbol{u})=\left[\operatorname{argmax} Q_{i}\left(o_{i}, u_{i}\right)\right]_{i=1}^{N}$ . If multiple optimal joint-actions exist in the right side of the “=”, RL for LDAS is difficult to train because multiple return-maximum peak landscapes may impact the convergence and exploitation of RL, so new action selection and training schemes are required. The social conventions, especially the position-based priority can introduce partial ordering and break ties. This priority ordering can help exploitation around the smallest lexicographic joint-action. More responses can refer to our revised paper.
>
> For the "Grammar+sundry things" part:
>
> The grammar things have now been revised and modified, and can refer to the corresponding section. The hyperthesis means that we should set the reward and discount factor in RL properly so that maximizing the return of the MDP $\leftrightarrow$ minimizing our real-world goal, say, the step to cover all shaded cells. In the appendix, we give an example of an improper setting and proper setting for CCPP MDP.
>
> For the "Writing" part:
>
> We now make clearer our fig2 and adds words to explain our motivation, please see the revision. Social conventions have nothing to do with a macroscopic view. We break a macro step (i.e., a decision event) of MDP into micro-steps. At each micro-step, an optimal joint action can be selected out one component by one component in micro-steps according to the conventions, besides of being selected out simultaneously, i.e., the "microscopically sequenceable" defined in RL language is that the joint action $\pi^*(s)=(a_1,a_2,...,a_n) $ where each component can be produced as $a_1=\pi^*_1(s),a_2=\pi^*_2(s|a1),a_3=\pi^*_3(s|a2),...$ and $\pi_i$ functions are related. By doing so, The action selection process breaks the exponential action space into linearly expressible space complexity. Moreover,  the action representation scheme and selection process will not lose the optimality of the action space and can select out the lexicographic-smallest optimal action vector.

---

> > ### Comment · Reviewer_3mod · 2022-08-06
> > **Still have trouble understanding the contributions of the paper, likely due to writing issues**
> >
> > Thanks to the authors for this response. I think this is a better paper (and that there's a good paper inside of the current paper, just one that still needs a lot of work on the writing to get to) though I do not believe I can improve my score to an accept as the contributions of the paper are still extremely unclear. I believe that this is a writing issue / insufficient clear demonstration of the ideas in the experiments. However, before getting to the issues I still have, I **highly recommend coloring the changes between papers to make it easy for reviewers to see what has changed in the paper.** This is just a useful practice and a friendly recommendation to the authors.
> >
> > **Issue 1:**
> > Putting that aside, I take issue with several pieces of the claim *"If multiple optimal joint-actions exist in the right side of the “=”, RL for LDAS is difficult to train because multiple return-maximum peak landscapes may impact the convergence and exploitation of RL, so new action selection and training schemes are required. The social conventions, especially the position-based priority can introduce partial ordering and break ties"*.
> >
> > My issue is that this claim is, as stated, an opinion and should instead be backed up with an experiment demonstrating clearly that this is a problem and that the proposed methods solve this issue rather than leading to improvements by solving some other issue.
> >
> > Besides this, there are other places in the paper where opinions are stated as facts.
> >
> > **Issue 2:**
> > The paper proposes to do something about microscopic sequenceability but only contains an explanation in the text. Since this is basically the key concept in the paper it should stand alone as a definition. Additionally, what it means to view the system `macroscopically` should be described more carefully.
> >
> > **Issue 3**:
> > Line 181 still contains a claim about "physical" meaning. I do not think the authors are using the term "physical" correctly; I think they mean that it has an intuitive meaning?
> >
> > **Issue 4**:
> > Is the contribution of this paper about tie-braking or about large discrete action spaces? I can't tell from the introduction or the motivation and the authors seem to flip-flop between the two. I'm assuming that the underlying argument is actually that multi-agent systems are inherently large action space due to exponential blow-up of the joint action space but that is not gotten across through the introduction.
> >
> > **Issue 5**:
> > What is the importance of `linearly expressible space complexity`? This term is not defined and simply introduced in one of the theorems.
> >
> > Some other small things:
> > - the axes in Figure 6 are too small and I am not convinced that there is a statistically significant difference between PCRL and DQN on 6a.
> > - in Figure 2b the text overlaps with one of the boxes
> > - amd ryzen should be capitalized as AMD Ryzen
> > - Table 1 could round the values to 1 decimal place without any significant loss of information. There should be more spacing around the text in the explanation.
> > - The axes in Fig. 3b should probably be adjusted

---

> > > ### Author Response · Authors · 2022-08-09
> > > **Authors new response**
> > >
> > > Thanks for the suggestions and help. We will do as recommended.
> > >
> > > For issue 1: The question is very good. Since we did not express it clearly, we are sorry for the misunderstanding. We will modify the corresponding parts of the manuscript to make the expression more accurate, clearer, and more backed up.
> > >
> > > For issue 2: The macroscopic view of PCRL treats a decision of the multi-agent system as one macro-step, and this macro-step is the decision step of the MDP: The planner maps the current state (including map state + agent state) to an optimal action vector whose first component is for the highest priority agent, etc. That is \begin{equation}
> > > 			\vec{a_t}=f(s_t).
> > > 		\end{equation} The macro-step microscopically consists of several micro-steps.
> > >
> > > For issue 3: Yes. Your suggestion is more accurate.
> > > For issue 4: The main contribution of this paper is about large discrete action spaces. We deal LDAS with microscopical sequencing. As multiple optimal sequences (i.e., action vectors) exist, we introduce priority to break ties, learning convergence to the lexicographic-smallest optimal action vector.
> > >
> > > For issue 5: Sorry for the unclearness. This means that the parameters for LDAS will not increase exponentially with the number of agents but linearly or remain constant.
> > >
> > > For other things: Thanks very much for pointing out these errors. We apologize for these grammatical problems and have corrected them based on the suggestions.

---

### Official Review · Reviewer_Kbn5 · 2022-07-13

**Rating:** 3
**Confidence:** 4
**Soundness:** 1 poor
**Presentation:** 1 poor
**Contribution:** 1 poor

**Summary:**

This paper looks to tackle multi-agent reinforcement learning problems operating in large-discrete-action spaces. The authors do so by introducing a priority convention into the RL loop. The priority convention results in generating a sequence of actions that are somehow better. The authors demonstrate the effectiveness of their algorithm by demonstrating experiments on various multi-robot problems.

**Questions:**

I do not think this paper in its current form meets the high bar for publication at NeuRIPS. The paper is weak in many areas from writing. novelty, actual methodology, experimentation.

I recommend the authors take a step back and re evaluate the proposal here and what introducing priorities does to the problem and if it even needs RL if one were to introduce a heuristic like priority convention.

**Ethics Review Area:**

["I don’t know"]

**Limitations:**

Please refer above.

**Strengths And Weaknesses:**

Strengths
1. The paper is thorough with its literature review.

Weaknesses
1. The paper is extremely poorly written and throws around a lot of words that carry no meaning scientifically. For example in the abstract alone "In this work, we propose to embed agreed priority conventions into reinforcement learning (PCRL) to directly tackle the microscopically sequencable multi-agent LDAS problems" To the best of my knowledge I have never heard the term microscopically sequencable (and after reading the paper I still do not know what the authors are trying to say with the term microscopically sequencable means in a rigid mathematical sense.) Further phrases such as "can help exploitation around the lexicographic-smallest optimal action vector" make no sense to me. Such phrases and terms occur in the paper at multiple places and either serve very little purpose or only stand to confuse the reviewer when not rigidly defined. The phrase cargmax or consistent argmax at a first glance only confuses the reader. I would very sincerely request the authors to overhaul Section 3 and define everything instead of vaguely throwing out terms that either mean nothing or only confuse the reader which is to say nothing of the poor choice of notation (writing out "one hot", ? in Line 198, writing out "length") that only makes things worse.
2. Now to come to the actual methodology devised in the paper here; the authors say nothing about what the actual priority convention is anywhere in the paper or in their experiments. How is the priority convention decided? How is it set. What are the parameters. How does it affect the training. In the experiments the robots all look identical and how does interchangeable convention set precedence to anything; i.e if Robot 1 is the same as Robot N then how does it matter if I switch priority from 1 to N and vice-versa.
3. Further, the paper itself doesn't say anything about what happens if you force in a priority convention/ie why do you still need RL. In most cases in multi-robot problems, setting a priority convention is akin to setting a heuristic and removes the need for any RL. One chooses multi-agent RL if there exists a hard to define objective function for each robot but there exists a global objective function we wish to optimize for. If one is able to discretize this and say R1 is more important than Rn, then it is fairly trivial (especially in the grid world situations) to design a controller without any RL or machine learning that simply optimizes for the R1 > Rn objective. If one thinks back about the problem in Fig 1c, if, there exists a priority convention for the 18 robots, you do not need a learning based solution and can simply write a dfs solution with some edges carrying more weights (corresponding to the priority convention) and have the optimal solution. In the case where you have stochasticity it still seems to me that one can get away without doing any RL and setup an objective function that accounts for the stochasticity.
4. Lastly, the experiments are extremely poorly designed. The proposed method needs to be compared with a relevant algorithm, By biasing the proposed algorithm and not providing the baseline DQN any reward based on the priority convention or information about the priority convention, the baseline is handicapped. Further, the experiments themselves are not compelling enough to determine that the proposed method here outperforms any standard non heuristic enabled multi-agent RL algorithm.

---

> ### Author Response · Authors · 2022-08-02
> **Author response**
>
> Thanks for the reviewer's comments and questions.
>
> For Q1: Some multi-agent problems are microscopically sequenceable, i.e., an optimal joint action can be selected out one component by one component in micro-steps according to the conventions, besides of being selected out simultaneously, i.e., the "microscopically sequenceable" defined in RL language is that the joint action $\pi^*(s)=(a_1,a_2,...,a_n) $ where each component can be produced as $a_1=\pi^*_1(s),a_2=\pi^*_2(s|a1),a_3=\pi^*_3(s|a2),...$ and $\pi_i$ functions are related. lexicographic order, cargmax and other inaccuracies all have been further clarified.
>
> For Q2:
> The agreed priority conventions for a specific multi-agent problem are prescribed by humans before the learning starts, and can be problem-specific designed. The priority conventions usually consist of (1) \textbf{Agent priority convention}: This is for determining the order of agents to decide action and breaking symmetries: Dynamically, at each decision step of the centralized system, the agents' priorities are determined by current states, say, the relative positions of agents. For example, in \fref{example}(c), the agents' priorities can be determined from left to right and upper to lower position, so the priorities (deciding order) of agents are $R_{1}\prec R_{2}...\prec R_{16}$. Here, $x\prec y$ means $x$ has a higher priority than $y$. If the priority is not position based but ID based, then the state space will have many mirror states. (2) \textbf{Agent ID convention}: This is for coinciding agents and breaking symmetries. Statically assign each agent a unique ID before learning to identify itself from others. When at least two agents coincide in the same position during the application, the smaller id agent has a higher priority. (3) \textbf{Action priority convention}: This is for breaking ties of equally optimal actions. The actions of an agent have prescribed priority, say, up action$\prec$left action$\prec$down action$\prec$right action, to break action ties if two actions both will bring about optimality, and to converge to one minimum lexicographic action vector. Hereafter, we define that two action vector $\vec{\alpha}\prec \vec{\beta}$ (or called "$\vec{\alpha}$ is \textbf{lexicographic smaller} than $\vec{\beta}$") if and only if $\exists i\in [1,N],\ \alpha_i\prec \beta_i$ and  $\forall j\in [1,i-1],\ \alpha_j=\beta_j$.
>
>
> The robots are identical and we need a precedence relationship to break symmetries and to lead to consistency for the optimal joint-action among the agents.  It doesn’t matter if you change the position priority, say the down most rightmost agent has the highest priority. Different priorities will get different $\pi^*(s)$, but the maximum return from the same current state is the same.
>
> For Q3:
> The reason we still need RL is because of the NP-completeness of the concerned problems. As the appendix pointed out, many planning paths for CCPP exist and to find the best one is complicated with exhaustive search or dfs search, so RL can help to optimize to (sub-)optimal via sample-based optimization. We based our RL on priority conventions. The objective of each agent is not defined, but the objective for those agents is to visit shaded cells the quickest.
>
> For Q4: Our method is centralized-training-centralized execution. It directly deals with large discrete action space ,so we were not meant to compare to the CTDE framework. We are now conducting experiments to comprehensively compare our work.

---

### Official Review · Reviewer_6W63 · 2022-07-25

**Rating:** 3
**Confidence:** 4
**Soundness:** 2 fair
**Presentation:** 2 fair
**Contribution:** 2 fair

**Summary:**

The paper aims to tackle the problem of large discrete action spaces in multi-agent MDPs (MMDPs) where the number of actions rises exponentially with the number of agents. The paper proposes to embed agreed priority conventions (agent priorities, agent ID, action priorities) into reinforcement learning (PCRL) to tackle large action spaces. The paper leverages RNNs to generate Q-values for the system one dimension of the action vector at a time sequentially (while maintaining priority conventions), and proposes an equality auxiliary loss to make sure that the V*(s) are equal for each RNN output corresponding to each action dimension. The paper claims that the priority conventions do not miss the optimal actions. The authors evaluate their proposed approach on coverage planning and pong domains.

**Questions:**

- Can the authors answer questions from the weakness/strengths section?

- Additional Questions

  - It seems like the different priority conventions (agent priorities, agent ID, action priorities) conventions are predefined before learning. Would it affect the convergence if conventions are changed? For example, if action priorities are changed, would it change the final solution? How would one define such priority conventions for large scale domains like Starcraft II [7]?

  - Line 235: the authors mentions that “PCRL network can successfully learn the conventions”. Aren’t the conventions already predefined for the domains?

**Limitations:**

Yes, the authors does mention the limitations of their work via a future work paragraph. Additionally, the authors should consider partial observable domains and possibly learning priority conventions.

**Strengths And Weaknesses:**

## General Comments
- The proposed approach seems sound and it is interesting to see authors approach match up to the performance of DQN with significantly lesser number of parameters required.

- The paper is a bit hard to understand at places with grammatical mistakes. Figure 2 makes it easier to understand the paper. It would be great if the authors could take some time to focus on the writing of the paper. Some points to consider:

  - For example:  “some researchers resorted to independent reinforcement learning (RL) regardless of other agents, but that will form loose cooperation and yield suboptimal solutions” → Independent MARL has shown to be a surprisingly good baseline in recently published papers in cooperative MARL. [6]
  - Some minor points:
    - Line 161: "where h_0 = C_0 is the featured and supplied state s" → I am not sure what the authors mean by featured and supplied state?
    - Line 166: ungaurenteed → unguaranteed; inconsitent → incosistent
    - Line 198: (??) reference missing

- The paper is missing several key citations from recent MARL literature which especially focuses on improving exploration in MARL large action spaces with role-based learning (RODE: [1]) as well as papers leveraging the centralised training decentralised execution (CTDE) framework (an intermediary between fully decentralised learning and centralised learning) [2,3,4,5]

- The idea of leveraging RNNs to predict one dimension of the action vector at a time seems to be very similar to the other works [7,8] with similar equality constraints as proposed by authors in Eq. 11. Can the authors comment on the novelty of their approach?

- The considered problem in the paper seems to be difficult from the perspective of a large number of agents (thereby large action spaces), and therefore the proposed approach seems to be only useful for scenarios with full observability and homogeneous agents which is not commonly observed in real-world multi-agent problems.

## Comments on Experimental Section

- The authors only compare their approach against the joint-action DQN based approach. There have been several recent works on tackling the large action spaces in MARL with role-based learning [1], and other CTDE based approaches [2,3,4,5] which can tackle large action spaces pretty effectively especially on the coverage and pong domains in the paper. The authors should compare against these approaches to fully showcase the effectiveness of their approach.

- The authors does not evaluate on any large-scale multi-agent RL domains like Starcraft II (SMAC) [7] (with some maps having 27 agents and each agent allowed to take 8 actions) which would help prove the efficacy of their method on large action space domains.

[1] Wang, Tonghan, et al. "Rode: Learning roles to decompose multi-agent tasks." ICLR 2020.

[2] Son, Kyunghwan, et al. "Qtran: Learning to factorize with transformation for cooperative multi-agent reinforcement learning." International conference on machine learning. PMLR, 2019.

[3] Mahajan, Anuj, et al. "Maven: Multi-agent variational exploration." Advances in Neural Information Processing Systems 32 (2019). APA

[4] Rashid, Tabish, et al. "Weighted qmix: Expanding monotonic value function factorisation for deep multi-agent reinforcement learning." Advances in neural information processing systems 33 (2020): 10199-10210. APA

[5] Son, Kyunghwan, et al. "QTRAN++: improved value transformation for cooperative multi-agent reinforcement learning." arXiv preprint arXiv:2006.12010 (2020).

[6] de Witt, Christian Schroeder, et al. "Is independent learning all you need in the starcraft multi-agent challenge?." arXiv preprint arXiv:2011.09533 (2020).

[7] Samvelyan, Mikayel, et al. "The starcraft multi-agent challenge." arXiv preprint arXiv:1902.04043 (2019).

---

> ### Author Response · Authors · 2022-08-02
> **Author response**
>
> Thanks for the reviewer's comments and questions.
>
> For the "General Comments" part:
>
> Thank you very much for your suggestion, and sorry that we have improperly written here.We have modified the original statement in the revised version. We want to express that independent reinforcement learning methods will face nonstationary problems in the multi-agent environment.
>
> Grammar things have all been carefully revised.
>
> Compared with the algorithm of CTDE learning paradigm(for example the Qmix,RODE), we proposed a multi-agent sequential decision algorithm inspired by social conventions and our algorithm does not need value decomposition constraints. In order to reflect the superiority of our method, we have been conducting experiments these days comparing PCRL with the existing CTDE algorithms in benchmark environments. We also want to note that our approach directly faces to the LDAS problem and optimize toward the lexicographic smallest optimal joint action. Our work is to help exploitation around the lexicographic smallest joint action. For the CTDE framework, some work needs the IGM principle, some uses factor mechanism, and some cannot tackle LDAS, so we cite some representative work of them in the related work section. These are the novelty and application-oriented settings of our work. Though not commonly observed but practicable.
>
> For the "Comments on Experimental Section" part: The paper now has the contribution of (1) Priority conventions and proofs: this paper proposes the concept of "microscopically sequenceable" to tackle LDAS multi-agent problems without missing the optimality, and can help exploitation around the lexicographic-smallest optimal action vector. (2) New schemes: it proposes new learning schemes that fully exploit the agreed conventions, such as auxiliary equality constraint, and neural network action selection schemes for PCRL. (3) Proof-of-concept practices: applying PCRL to tackle seemingly different problems, including $10^{10}$ magnitude LDAS multi-agent path planning problems and the pixel cooperative task, and achieves 20\% fewer steps and competitive performance.” In order to further highlight the advantages of our method, we have established problems such as 27m_ vs_ 30m StarCraft game scene as a relative-position-based microscopically sequenceable problem and have been conducting experiments to verify the algorithm on a larger action space problem.
>
> For the "Questions" part:
> The agreed priority conventions for a specific multi-agent problem are prescribed by humans before the learning starts, and can be problem-specific designed. The priority conventions usually consist of (1) \textbf{Agent priority convention}: This is for determining the order of agents to decide action and breaking symmetries: Dynamically, at each decision step of the centralized system, the agents' priorities are determined by current states, say, the relative positions of agents. For example, in \fref{example}(c), the agents' priorities can be determined from left to right and upper to lower position, so the priorities (deciding order) of agents are $R_{1}\prec R_{2}...\prec R_{16}$. Here, $x\prec y$ means $x$ has a higher priority than $y$. If the priority is not position based but ID based, then the state space will have many mirror states. (2) \textbf{Agent ID convention}: This is for coinciding agents and breaking symmetries. Statically assign each agent a unique ID before learning to identify itself from others. When at least two agents coincide in the same position during the application, the smaller id agent has a higher priority. (3) \textbf{Action priority convention}: This is for breaking ties of equally optimal actions. The actions of an agent have prescribed priority, say, up action$\prec$left action$\prec$down action$\prec$right action, to break action ties if two actions both will bring about optimality, and to converge to one minimum lexicographic action vector. Hereafter, we define that two action vector $\vec{\alpha}\prec \vec{\beta}$ (or called "$\vec{\alpha}$ is \textbf{lexicographic smaller} than $\vec{\beta}$") if and only if $\exists i\in [1,N],\ \alpha_i\prec \beta_i$ and  $\forall j\in [1,i-1],\ \alpha_j=\beta_j$.
>
> Different priorities will get different $\pi^*(s)$, but the maximum return from the same current state  is the same.

---

### Meta-Review · Area_Chair_LnpG · 2022-08-21

**Recommendation:** Reject
**Confidence:** Certain

**Metareview:**

While the ideas in this paper are promising, there are issues with the paper's presentation and experimental results. The paper needs to be (further) updated to clarify the proposed method and discuss additional related work. More extensive experimental results are also needed to show the benefits of the proposed approach.

**Award:**

No

---

### Decision · Program_Chairs · 2022-09-14

Reject